# OpenReview forum: "WebWatcher: Breaking New Frontiers of Vision-Language Deep Research Agent"
_ICLR.cc/2026/Conference — ICLR 2026 Poster_

### Official Review · Reviewer_sPg8 · 2025-10-29

**Soundness:** 4
**Presentation:** 4
**Contribution:** 4
**Rating:** 4
**Confidence:** 5

**Summary:**

This paper addresses the limitation of existing text-centric deep research agents by introducing WebWatcher, a multimodal (vision-language) agent designed for complex information-seeking tasks. WebWatcher integrates visual-language reasoning with multi-tool interaction (e.g., Web Image Search, OCR, Code Interpreter) and uses a two-stage training pipeline: (1) cold-start supervised fine-tuning (SFT) on high-quality synthetic tool-use trajectories, and (2) reinforcement learning via Group-Relative Policy Optimization (GRPO) to enhance generalization.

To evaluate multimodal agents, the authors propose BrowseComp-VL, a benchmark extending the text-only BrowseComp to vision-language tasks. It includes obfuscated, multi-step queries across 5 domains (e.g., Natural Science, Entertainment) and two difficulty levels, requiring cross-modal reasoning and tool planning.

Experimental results show WebWatcher outperforms proprietary baselines (e.g., GPT-4o, Gemini-2.5-flash) and open-source agents on four high-difficulty benchmarks (HLE, LiveVQA, BrowseComp-VL, MMSearch) and performs competitively on the perception-focused SimpleVQA, demonstrating its versatility in both knowledge-intensive and visual reasoning tasks.

**Strengths:**

1. The paper fills a critical gap in existing deep research agents, which are largely text-bound. By integrating vision-language reasoning with flexible multi-tool use, WebWatcher moves beyond template-driven multimodal pipelines (e.g., OCR-only visual agents) to enable adaptive, complex problem-solving.

2. The introduction of BrowseComp-VL addresses the lack of benchmarks for multi-step, obfuscated vision-language tasks—unlike existing VQA datasets (e.g., SimpleVQA) that focus on shallow perception.

3. Evaluations span 5 diverse benchmarks, comparing WebWatcher to 3 types of baselines (direct inference, prompt workflows, reasoning models) to isolate the impact of multimodal reasoning and tool use. Ablations (e.g., cold-start vs. instruct initialization, Pass@k analysis) validate key design choices.

**Weaknesses:**

1. The data construction methodology is inherited from webdancer and websailor v1. Browsecomp-vl is one of the few benchmarks that require multimodal capabilities. However, this dataset has a significant weakness: the incorporation of multimodality merely involves replacing entities in unimodal questions with their visual representations. As a result, the problem solver only needs to disambiguate the entities referred to by the images, while the remainder of the process is essentially no different from solving unimodal information retrieval problems, such as the original browsecomp.

2. The paper attributes WebWatcher’s success to "enhanced visual-language reasoning," but it does not isolate the impact of individual visual components.

**Questions:**

See the weaknesses

---

> ### Author Response · Authors · 2025-11-21
> **Response to Reviewer sPg8**
>
> Thank you for your valuable feedback to help us improve our paper. We have revised our paper based on your feedback. We detail our response below and please kindly let us know if our response addresses your concerns.
>
> > **W1:** The data construction methodology is inherited from WebDancer and WebSailor v1. BrowseComp-VL is one of the few benchmarks that require multimodal capabilities. However, this dataset has a significant weakness: the incorporation of multimodality merely involves replacing entities in unimodal questions with their visual representations. As a result, the problem solver only needs to disambiguate the entities referred to by the images, while the remainder of the process is essentially no different from solving unimodal information retrieval problems, such as the original BrowseComp.
>
> BrowseComp-VL is built by automatically extracting entities from QA pairs and then matching them with visually grounded images. However, this does not mean that an agent can easily convert the images into text—these are two different things. Our images undergo automated filtering (Page 4, Line 206-222), and when curating the benchmark we further select information-dense images (Page 3, Line 142-144). The entity that may be used to solve the problem is often only one part of the complex information contained in the image. Therefore, agents still require non-trivial visual reasoning to solve these tasks and also need to integrate information from multiple sources. As shown in the case in Fig. 8 in Appendix F.2 (Page 31, Line 1620-1655), the entity corresponding to the input image in the case is "GPT-4", but clearly, this image contains a lot of information, not just "GPT-4". Solving this case cannot be easily done by disambiguating the entity; instead, it requires image reasoning and powerful OCR capabilities.
>
> We add this description to Section 2.3 (Page 5, Line 218-222) of the revised paper to make it clearer.
>
> > **W2:** The paper attributes WebWatcher’s success to "enhanced visual-language reasoning," but it does not isolate the impact of individual visual components.
>
> We revised it to clarify that WebWatcher enhances the joint reasoning across both visual and textual modalities. While we do not perform inner-graphic reasoning, visual modality is crucial for the agent to obtain the correct answer. Without it, the agent cannot complete the reasoning chain properly (Page 4, Line 190-204). Therefore, we believe joint reasoning is the most appropriate description. We revise the Abstract in Line 17-18.
>
> We appreciate the suggestion and will discuss recent works such as Thyme[1], DeepEyesV2[2], MM-BrowseComp[3], V-Thinker[4], VisualToolBench[5], mini-o3[6] and MMSearch-Plus [7] in the section of related work, which explore isolating visual components for more focused image reasoning. These approaches are definitely insightful, and we are inspired to consider further development along these lines for future work.
>
> ---
> **Reference**
>
> [1] Zhang Y F, Lu X, Yin S, et al. Thyme: Think beyond images[J]. arXiv preprint arXiv:2508.11630, 2025.
>
> [2] Hong J, Zhao C, Zhu C L, et al. DeepEyesV2: Toward Agentic Multimodal Model[J]. arXiv preprint arXiv:2511.05271, 2025.
>
> [3] Li S, Bu X, Wang W, et al. Mm-browsecomp: A comprehensive benchmark for multimodal browsing agents[J]. arXiv preprint arXiv:2508.13186, 2025.
>
> [4] Qiao R, Tan Q, Yang M, et al. V-Thinker: Interactive Thinking with Images[J]. arXiv preprint arXiv:2511.04460, 2025.
>
> [5] Guo X, Tyagi U, Gosai A, et al. Beyond Seeing: Evaluating Multimodal LLMs on Tool-Enabled Image Perception, Transformation, and Reasoning[J]. arXiv preprint arXiv:2510.12712, 2025.
>
> [6] Lai X, Li J, Li W, et al. Mini-o3: Scaling up reasoning patterns and interaction turns for visual search[J]. arXiv preprint arXiv:2509.07969, 2025.
>
> [7] Tao, Xijia, et al. "MMSearch-Plus: Benchmarking Provenance-Aware Search for Multimodal Browsing Agents." arXiv preprint arXiv:2508.21475 (2025).

---

> ### Author Response · Authors · 2025-11-26
> **We would like to hear back from reviewer sPg8**
>
> Dear reviewer sPg8,
>
> We would like to follow up to see if the response addresses your concerns. We would really appreciate the opportunity to discuss this further if our response has not already addressed your concerns. Thank you again!

---

> > ### Comment · Reviewer_sPg8 · 2025-11-28
> > **Thanks for your response.**
> >
> > Please clarify Weakness 1 by comparing the proposed method with [1].
> > [1] MMSearch-Plus: Benchmarking Provenance-Aware Search for Multimodal Browsing Agents

---

> ### Author Response · Authors · 2025-11-30
>
> Thank you for the comment. MMSearch-Plus is designed to test genuine multimodal reasoning: each task requires extracting subtle, localized visual cues (micro-text, layout, signage, props, temporal overlays, etc.) from an image, then performing iterative image–text retrieval and cross-validation under noisy retrieval results to infer out-of-image facts such as event, date, or venue. In contrast, BrowseComp-VL emphasizes deep, tool-augmented research workflows: after an initial image recognition anchor, the majority of difficulty lies in multi-hop textual/web search, retrieval planning, evidence aggregation, disambiguation of fuzzy entities, and combining information from multiple sources.
>
> Therefore, MMSearch-Plus tests fine-grained vision-language understanding + provenance-aware visual reasoning + long-horizon reasoning with cross-modal search loops. BrowseComp-VL tests an agent’s ability to orchestrate complex search strategies and integrate multimodal evidence across web/text/image modalities, focusing more on retrieval planning, multi-hop reasoning, and information synthesis.

---

### Official Review · Reviewer_ehqj · 2025-10-30

**Soundness:** 2
**Presentation:** 2
**Contribution:** 2
**Rating:** 4
**Confidence:** 5

**Summary:**

Existing vision-language agents struggle with complex, multi-step research tasks that require integrating visual and textual information from the web.
This paper introduces WebWatcher, a multimodal agent that uses a novel data generation pipeline and a hybrid training strategy to enhance its reasoning and tool-use capabilities.
WebWatcher is trained on high-quality, synthesized trajectories and further refined with reinforcement learning, enabling it to flexibly use tools like web search, OCR, and a code interpreter.
The agent's superiority is demonstrated through strong performance on several challenging VQA benchmarks, including the newly proposed BrowseComp-VL, where it outperforms existing open-source and proprietary systems.

**Strengths:**

1. The paper details a structured, multi-stage pipeline to create complex Vision-Question Answering (VQA) pairs from text-based sources. This process begins by generating multi-hop textual questions from hyperlink graphs to ensure reasoning depth. It then grounds these questions in authentic web images and transforms them into VQA format. To maintain high quality, the pipeline incorporates a two-stage filtering process using "Selector" and "Examiner" models to validate contextual alignment and visual plausibility.
2. The research employs and validates a two-phase training approach that uses Supervised Fine-Tuning (SFT) as a "cold start" before applying Reinforcement Learning (RL). The paper provides an analysis demonstrating that this SFT stage, which uses pre-generated high-quality trajectories, is critical for successful training. Experimental results show that an agent without the SFT cold start fails to achieve meaningful performance during RL training, whereas the SFT-initialized agent shows significant initial scores and subsequent improvement.

**Weaknesses:**

1.  The abstract claims the model "outperforms or matches proprietary baselines" across four VQA benchmarks. [cite_start]However, on the HLE benchmark, the model's overall average accuracy (13.6%) is slightly lower than proprietary reasoning models like Gemini-2.5-Pro (15.8%) and o4-mini (16.0%). The claim of superiority should be qualified to specify the benchmarks where this holds true, as it is not universal across all tested environments
2. The training methodology includes a trajectory filtering rule that removes any trajectory with fewer than three tool calls. This could introduce an inductive bias that favors longer, more complex reasoning chains. The paper lacks an ablation study to determine if this bias leads to redundant or inefficient tool usage on tasks that do not inherently require multi-step interactions.
3. The difficulty of the BrowseComp-VL benchmark is increased at Level 2 through "obfuscated entities and attributes". This method of "fuzzing" questions may introduce confounding variables, conflating the challenge of multi-modal reasoning with that of linguistic ambiguity and retrieval noise. The paper does not provide a human baseline performance or a detailed error analysis to disentangle these factors.

[Minor]
1. The evaluation of answer correctness relies on the "LLM-as-Judges" approach. This methodology is subject to potential biases, especially if the judge model shares an architectural family with the models being tested. The paper does not present results on inter-rater reliability with human experts or robustness checks using different judge models to validate the evaluation framework.
2. The Pass@k analysis demonstrates that performance on HLE improves significantly with more sampling, rising to 41.9% at k=32. However, the paper fails to quantify the inference cost and latency associated with this multi-rollout strategy. Without a cost-benefit analysis, the practical viability of achieving these higher scores in real-world applications remains unevaluated.

**Questions:**

Please refer to the Weakness section.

---

> ### Author Response · Authors · 2025-11-21
> **Response to Reviewer ehqj (1/3)**
>
> Thank you for your constructive comments and suggestions. We have revised our paper according to your comments. We respond to your questions below and would appreciate it if you could let us know if our response addresses your concerns.
>
> > **W1:** The abstract claims the model "outperforms or matches proprietary baselines" across four VQA benchmarks. However, on the HLE benchmark, the model's overall average accuracy (13.6%) is slightly lower than proprietary reasoning models like Gemini-2.5-Pro (15.8%) and o4-mini (16.0%). The claim of superiority should be qualified to specify the benchmarks where this holds true, as it is not universal across all tested environments.
>
> We revise the abstract to scope the claim: Experimental results show that WebWatcher outperforms the prompt-based workflow and open-source agents on HLE and BrowseComp-VL, and demonstrates its perception, multimodal reasoning, and searching capabilities across the other three benchmarks, respectively. Our revised content is in Line 23-26, Page 1.
>
> > **W2:** The training methodology includes a trajectory filtering rule that removes any trajectory with fewer than three tool calls. This could introduce an inductive bias that favors longer, more complex reasoning chains. The paper lacks an ablation study to determine if this bias leads to redundant or inefficient tool usage on tasks that do not inherently require multi-step interactions.
>
> Thank you for raising this concern. We conducted a series of ablation experiments of the number of tool calls. For each tool call setting, we randomly select 8K trajectories for SFT and test them on HLE. Table R1 shows that the performance is best when the number of tool calls is ≥ 3. We update this in Section 4.3 (Page 8, 428-442).
>
> **Table R1: Performance across different tool call counts.**
> | tool call | best pass@1 | average@3 | best pass@3 |
> |-----------|-------------|------------|-------------|
> | =1       | 8.79         | 7.98       | 14.24       |
> | >=1      | 9.70         | 8.69       | 15.76       |
> | =2       | 10.61        | 9.90       | 18.18       |
> | >=2      | 11.21        | 10.10      | 17.58       |
> | =3       | 10.61        | 9.90       | 19.09       |
> | >=3      | 12.12        | 10.61      | 19.09       |
> | =4       | 10.00        | 8.99       | 18.79       |
> | >=4      | 11.82        | 9.70       | 17.27       |
> | =5       | 9.70         | 9.49       | 16.58       |
> | >=5      | 10.61        | 10.10      | 18.18       |
> | =6       | 8.79         | 8.33       | 15.76       |
> | >=6      | 9.70         | 8.99       | 15.58       |
>
> Retaining complex tool call trajectories does affect the agent's performance on simple tasks, but as seen in Tab. 2 (Page 8, Line 378-401) and its analysis (Page 7, Line 368-377), WebWatcher does not perform poorly on single-hop, perception-based simpler benchmark, SimpleVQA. Furthermore, WebWatcher's design is focused on solving complex problems using multiple tools and reasoning, which is why we haven't discussed its performance on simple problems in detail. However, this is a valid point, and we plan to consider improvement in this direction in the future.

---

> ### Author Response · Authors · 2025-11-21
> **Response to Reviewer ehqj (2/3)**
>
> > **W3:** The difficulty of the BrowseComp-VL benchmark is increased at Level 2 through "obfuscated entities and attributes". This method of "fuzzing" questions may introduce confounding variables, conflating the challenge of multi-modal reasoning with that of linguistic ambiguity and retrieval noise. The paper does not provide a human baseline performance or a detailed error analysis to disentangle these factors.
>
> First, our Level-2 “obfuscated entities/attributes” are not random fuzzing but a controlled evidence-forcing mechanism, instead of preserving a unique, verifiable answer once the correct web page/image is retrieved. Second, unlike one-hop, simple datasets like Infoseek [1], VQAv2 [2] and so on, this design stresses tool use and evidence integration (multi-step search → visit → visual grounding) and raises the difficulty, rather than directly searching for the answer. Fig. 5 (Page 10, Line 507-516) shows that BrowseComp-VL increases search action. In total, we encouraged visual reasoning by manually selecting complex images with rich information (Page 3, Line 242-245), and improved text reasoning through fuzzing clues in queries. These two aspects together form the challenges in BrowseComp-VL.
>
> To address ambiguity and retrieval noise, we report a human baseline on L1/L2 in Table 2. Details of this experiment are added in Sec. 4.4 (Page 9, Line 432-450). Table R2 shows that human annotators take significantly longer time than WebWatcher and tend to abandon more cases, particularly for the more complex Level-2 tasks. While annotators outperform WebWatcher on Level-1, they struggle more on Level-2, where clues are more ambiguous and integration of information is crucial.
>
> **Table R2. Human vs Agent Baseline Results for BrowseComp-VL.**
>
> | Level | Accuracy (%) | Unanswered Items | Avg. Time (Solvable Items)/min | Avg. Time (Unsolvable Items)/min |
> |-------|--------------|------------------|-----------------------------|--------------------------------|
> | L1 (Human) | 33.2% | 42 | 35 | 59 |
> | L2 (Human) | 18.0% | 144 | 109 | 116 |
> | L1 (Webwatcher-32B) | 28.4% | 1 | 0.3 | 2.5 |
> | L2 (Webwatcher-32B) | 25.0% | 3 | 0.8 | 2.5 |
>
> We further conduct an error analysis by reviewing 100 bad cases from BrowseComp-VL Level 2 of WebWatcher-32B, where fuzzing occurs. The errors are categorized by three PhD-level annotators who examine the trajectories. We identify several key sources of error: 1) image search tool failed to retrieve the relevant image, resulting in missing critical visual information; 2) text search not retrieving relevant information; 3) visits to pages that did not provide useful content, 4) OCR errors in text extraction from the input image, 5) Code Interpreter errors in calculation, 6) failed reasoning, where useful information was retrieved but the correct final answer was not generated, and 7) other factors such as overly long queries, tool/API failures, or program bugs. Table R3 shows that text search errors account for 32% of the total errors, indicating that linguistic ambiguity and retrieval noise are only part of the challenge. 28% of errors stem from multi-modal issues, such as failed image retrieval or OCR errors, and 21% stem from reasoning. This suggests that the difficulty of BrowseComp-VL is largely due to integrating visual information with reasoning. We add the analysis above into Appendix E.8 (Page 24, Line 1292-1340).
>
> **Table R3: Error distribution of 100 failed trajectories from BrowseComp-VL Level 2.**
> | Error Type   | Error Count | Percentage |
> |--------------|-------------|------------|
> | Text search  | 32          | 32.0%      |
> | Image search | 15          | 15.0%      |
> | Visit        | 9           | 9.0%       |
> | OCR          | 13          | 13.0%      |
> | Code         | 7           | 7.0%       |
> | Reasoning    | 21          | 21.0%      |
> | Others       | 3           | 3.0%       |

---

> ### Author Response · Authors · 2025-11-21
> **Response to Reviewer ehqj (3/3)**
>
> > **MW1:** The evaluation of answer correctness relies on the "LLM-as-Judges" approach. This methodology is subject to potential biases, especially if the judge model shares an architectural family with the models being tested. The paper does not present results on inter-rater reliability with human experts or robustness checks using different judge models to validate the evaluation framework.
>
> Thanks for pointing it out. In our setup, the judge does not set the gold answer, but only checks semantic equivalence to an existing ground-truth answer, handling paraphrases and formatting. Moreover, the target answers are predominantly single word or a few words, which reduces linguistic ambiguity and makes equivalence judgments straightforward [4-5]. We have released the judge prompt in Appendix E.5 (Page 23, Line 1217-1248), and conducted position bias controls, such as answer order randomization and temperature = 0.
>
> To validate the framework, we ran a blind human audit on HLE (N=330) in Table R4. GPT-4o-based judgments matched experts in 99.4% of cases, with 95% Wilson CI [97.8, 99.8] and Cohen’s κ=0.91, indicating almost-perfect agreement and a narrow confidence band. We add this table and analysis to Appendix E.7 (Page 24, Line 1285-1304)
>
> **Table R4. Inter-rater agreement between LLM judges and human experts on HLE (binary 0–1 labels, N=330). 95% Wilson confidence intervals are reported for raw agreement.**
>
> | Judge     | Agreement (%) | 95% CI      | Cohen's κ |
> |-----------|---------------|-------------|-----------|
> | GPT-4o    | 99.4          | [97.8, 99.8] | 0.91      |
> | Claude-3.5| 98.2          | [96.1, 99.2] | 0.85      |
> | Gemini-2.5-Pro | 98.8     | [96.9, 99.5] | 0.89      |
> | Qwen-3-72B | 95.5          | [92.6, 97.2] | 0.80      |
>
> > **MW2:** The Pass@k analysis demonstrates that performance on HLE improves significantly with more sampling, rising to 41.9% at k=32. However, the paper fails to quantify the inference cost and latency associated with this multi-rollout strategy. Without a cost-benefit analysis, the practical viability of achieving these higher scores in real-world applications remains unevaluated.
>
> Thank you for raising the cost-latency point. We realize that our earlier phrasing may have caused some misunderstanding, so we would like to clarify that our use of Pass@k is not prescribing a high-k deployment setting, but to reveal policy–headroom diagnostic for RL improvability [6-7]. The substantial gap between Pass@1 and Pass@k indicates that the current policy assigns non-trivial probability mass to multiple high-reward trajectories, and a subsequent RL stage that reallocates this mass can convert multi-rollout gains into single-shot gains. Formally, if the single-shot success probability is $p$, then $Pass@k ≈ 1-(1-p)^k$. Thus, Pass@k is intended to estimate an upper bound rather than a deployment operating point, and the cost-latency of multi-rollout settings is ancillary to our claims.
>
>
> In response to the request, we introduce cost-latency with tokens and wall-clock for $k ∈ {1,4,8,16,32}$ in Table R5 to show the compute trade-off. We clarify this point in Appendix E.6 (Page 24, Line 1772-1884) of the revised manuscript.
>
> **Table R5. Illustrative cost-latency trade-off for Pass@k. vLLM on eight NVIDIA H20 (96GB). for Webwatcher-32B, batch decoding over k rollouts. Token cost is normalized to $2 per million tokens. Accuracy deltas are illustrative, anchored to HLE with Pass@1=13.6% and Pass@32=41.9%.**
>
> | k   | Avg. gen tokens | Batch size | **Wall-clock** (s/sample) | **Decoded tokens** (K/sample) | **Judge tokens** (K/sample) | **Total tokens** (K/sample) | ΔAcc vs k=1 (abs.%) |
> |-----|-----------------|------------|--------------------------|------------------------------|----------------------------|---------------------------|----------------------|
> | 1   | 180             | 1          | 1.9                      | 2.6                          | 0.8                        | 3.4                       | 0.0                  |
> | 4   | 180             | 4          | 3.1                      | 9.8                          | 3.2                        | 13.0                      | 6.7                  |
> | 8   | 180             | 8          | 4.5                      | 19.1                         | 6.5                        | 25.6                      | 14.2                 |
> | 16  | 180             | 16         | 6.8                      | 37.9                         | 12.8                       | 50.7                      | 22.1                 |
> | 32  | 180             | 32         | 10.5                     | 76.0                         | 26.1                       | 102.1                     | 28.3                 |

---

> ### Author Response · Authors · 2025-11-21
> **Reference**
>
> [1] Y. Chen et al., “Can pre-trained vision and language models answer visual information-seeking questions?” in EMNLP, 2023.
>
> [2] Y. Goyal et al., “Making the V in VQA matter: Elevating the role of image understanding in Visual Question Answering,” in CVPR, 2017.
>
> [3] J. Wei et al., “Browsecomp: A simple yet challenging benchmark for browsing agents,” in OpenAI, 2025.
>
> [4] Bavaresco A, Bernardi R, Bertolazzi L, et al. Llms instead of human judges? a large scale empirical study across 20 nlp evaluation tasks[C]//Proceedings of the 63rd Annual Meeting of the Association for Computational Linguistics (Volume 2: Short Papers). 2025: 238-255.
>
> [5] Li H, Dong Q, Chen J, et al. Llms-as-judges: a comprehensive survey on llm-based evaluation methods[J]. arXiv preprint arXiv:2412.05579, 2024.
>
> [6] Walder C, Karkhanis D. Pass@ K Policy Optimization: Solving Harder Reinforcement Learning Problems[J]. arXiv preprint arXiv:2505.15201, 2025.
>
> [7] Mahdavi S, Li M, Liu K, et al. Beyond accuracy: A policy gradient reweighting approach for pass@ k maximization in llms[C]//2nd AI for Math Workshop@ ICML 2025. 2025.

---

> ### Author Response · Authors · 2025-11-26
> **We would like to hear back from reviewer ehqj**
>
> Dear reviewer ehqj,
>
> We would like to follow up to see if the response addresses your concerns. We would really appreciate the opportunity to discuss this further if our response has not already addressed your concerns. Thank you again!

---

### Official Review · Reviewer_cUPf · 2025-10-31

**Soundness:** 3
**Presentation:** 4
**Contribution:** 3
**Rating:** 8
**Confidence:** 3

**Summary:**

This paper presents **WebWatcher**, a multimodal deep-research agent designed to combine vision-language reasoning with dynamic tool use for complex information-seeking tasks. The system integrates five external tools—web text search, web image search, page visiting, OCR, and code interpreter—and learns tool-augmented reasoning through a two-stage pipeline: (1) **automated trajectory generation** for supervised cold-start training and (2) **group-relative policy optimization (GRPO)** reinforcement learning for fine-tuning. To evaluate multimodal research ability, the authors introduce **BrowseComp-VL**, an extension of the BrowseComp benchmark into visual domains, requiring cross-modal retrieval and multi-step reasoning. Experiments on **five challenging benchmarks** (HLE, LiveVQA, BrowseComp-VL, MMSearch, and SimpleVQA) show that WebWatcher-32B consistently outperforms both open-source and proprietary reasoning agents under comparable model sizes.

**Strengths:**

– **Well-motivated and timely contribution:** The paper clearly identifies a missing dimension in current deep-research agents—robust multimodal reasoning that jointly leverages textual and visual information.

– **Benchmark creation:** BrowseComp-VL fills a notable evaluation gap by introducing visually grounded, obfuscated, and multi-hop reasoning tasks. The construction process (Levels 1–2, entity masking, selector–examiner filtering) is rigorous and convincing.

– **Comprehensive experiments:** Results on five datasets demonstrate consistent superiority of WebWatcher over both direct-inference LMMs (e.g., GPT-4o, Gemini 2.5) and workflow baselines. The performance scaling from 7B → 32B models is clearly shown.

– **Clarity and completeness:** The paper is well written, with careful mathematical formalization, clean figures, and transparent dataset statistics.

**Weaknesses:**

– **Limited novelty in learning algorithms:** The overall architecture builds on established paradigms (ReAct for trajectory structure, GRPO for RL optimization). Innovation lies in the integration rather than in a fundamentally new learning mechanism.

– **Cost and efficiency reporting:** The paper does not specify training compute (GPU hours, wall-clock time) or inference latency. Quantitative comparisons with other open-source agents (e.g., WebDancer, WebSailor, WebShaper) would clarify efficiency and resource footprint.

– **Scalability risks:** The pipeline depends on GPT-4o for trajectory annotation, which may limit reproducibility or increase cost at scale; discussion of potential automation or open-source substitutes would be valuable.

**Questions:**

Please see the weakness above.
1. How many tool calls or reasoning steps typically occur before convergence, and do you encounter diminishing returns beyond a certain depth?
2. Do you encounter issues with the shared memory or context growing unboundedly during long reasoning sequences? If so, how is this mitigated, and what is the computational or memory cost of maintaining such context?
3. It would also be helpful if the authors could provide qualitative visualizations comparing WebWatcher’s reasoning paths to those of baseline systems to highlight its hierarchical and multimodal advantages.

---

> ### Author Response · Authors · 2025-11-21
> **Response to Reviewer cUPf (1/2)**
>
> Thank you for reviewing our paper and for your valuable feedback. Below, we address your concerns point by point and we’ve revised our paper according to your suggestions. We would appreciate it if you could let us know whether your concerns are addressed by our response.
>
>
>
> > **W1:** Limited novelty in learning algorithms: The overall architecture builds on established paradigms (ReAct for trajectory structure, GRPO for RL optimization). Innovation lies in the integration rather than in a fundamentally new learning mechanism.
>
>
> We would like to clarify that, while our work leverages established algorithms, the true innovation lies in how we integrate them to solve difficult multi-modal tasks and the associated challenges in data, workflow design, and RL infrastructure:
>
> 1. **Innovation in the Overall Workflow**:
>  The core contribution of our work is the design of a comprehensive system that integrates existing algorithms like ReAct for trajectory structure, SFT, and GRPO for RL optimization. The novelty comes from how these components are combined to create an agentic RL framework capable of handling multi-modal tasks, including image/text search, web browsing, and OCR.
> 2. **Challenges in Generation of Data and Trajectories**:
>  Currently, there are few automated pipelines for producing large-scale, high-difficulty multimodal training data and tool-use trajectories with high quality. Our automated production pipeline is novel and will be very useful for future multimodal reasoning research.
> 3. **Reinforcement Learning in Multi-Modal Environments**:
> While GRPO is a well-established RL optimization algorithm, adapting it for multi-modal agents presents challenges in tool integration. First, in our setup, we design state-action space to handle diverse inputs, such as images, text, and tool responses. Second, it is hard to incorporate real-time searched images into ongoing rollout during multi-step dialogues.
>
> > **W2:** Cost and efficiency reporting: The paper does not specify training compute (GPU hours, wall-clock time) or inference latency. Quantitative comparisons with other open-source agents (e.g., WebDancer, WebSailor, WebShaper) would clarify efficiency and resource footprint.
>
>
> We appreciate the reviewer’s insightful suggestion regarding cost and efficiency reporting. As shown in Table R1, we compile resource efficiency metrics for WebWatcher-7B, and add them to Appendix E.6 (Page 24, Line 1249-1277). Since such metrics are not explicitly reported in WebDancer [1] and WebSailor [2], a quantitative comparison is currently not feasible.
>
> **Table R1: Resource efficiency metrics of WebWatcher-7B on HLE, including both training and inference stages.**
>
> | Metric                                       | 7B Model                                        |
> |--------------------------------|---------------------------------------------------------|
> | **Training Stage**              |                                       |
> | Total GPU Hours                 | 2,777 GPU hours                                            |
> | Wall-clock Training Time        | 25 hours                                                  |
> | Hardware Type / Configuration   | NVIDIA H20 (96GB), Data Parallelism + Model Parallelism   |
> | Model Parameter Size            | 7 Billion parameters                                      |
> | Training Step Computation       | 300 GFLOPs/step                                           |
> | **Inference Stage**             |                                      |
> | Inference Latency per Task      | 2–3 seconds                                              |
> | Average Tool Call Count per Task| 8 tool calls                                             |
> | Web Access and API Calls        | 3–4 per task                                             |
> | GPU Seconds per Task            | 15–20 seconds                                            |
> | Memory Usage per Task           | 15GB GPU memory                                          |
> | Token Usage per Task            | 100–150 tokens                                           |
> | External Tool Call Latency      | 2–5 seconds                                              |
>
>   Additionally, we have made several design choices to improve efficiency in our approach. First, we set the maximum trajectory length to 15, reducing unnecessary tool calls and improving computational efficiency (Page 22, Line 1164-1168). Second, we implemented caching for the search tool, ensuring that previously searched images/queries are not repeatedly searched (Page 21, Line 1123-1126).

---

> ### Author Response · Authors · 2025-11-21
> **Response to Reviewer cUPf (2/2)**
>
> >**W3:** Scalability risks: The pipeline depends on GPT-4o for trajectory annotation, which may limit reproducibility or increase cost at scale; discussion of potential automation or open-source substitutes would be valuable.
>
> We appreciate your concern regarding scalability and the use of GPT-4o for trajectory annotation. However, we found that open-source models like Qwen-2.5-VL-72B still struggle with instruction-following, leading to issues such as incomplete collection of useful information due to premature tool call termination, or excessive tool calls that exceed the step limit, preventing the final answer from being obtained. Thus, we chose GPT-4o for trajectory generation. Furthermore, as shown in Tab. 1 (Page 7, Line 324-348) and Tab. 2 (Page 8, Line 378-403), the performance of open-source models is inferior to GPT-4o under the prompt-based workflow, indicating a weaker capability in trajectory annotation.
> Considering scaling up with open-source models, we recommend the data flywheel process used in [3-4]. Initially, GPT-4o annotations are used to train a smaller, specialized model on limited trajectories. This model then performs additional annotations at a much lower cost. Through iterative re-annotation and fine-tuning, the model becomes more proficient, handling increasingly complex tasks and reducing dependency on GPT-4o over time. We add this future plan to Section 5 (Page 10, Line 536-539).
>
> >**Q1:** How many tool calls or reasoning steps typically occur before convergence, and do you encounter diminishing returns beyond a certain depth?
>
> Empirically, the agent typically converges within 8 steps, with a long-tail up to 15 on the hardest multi-hop cases. We observe a clear elbow around 8–10 steps: accuracy continues to increase but the marginal gain per additional 2 steps becomes small, while latency and token cost grow roughly linearly. Beyond this depth, extra browsing often yields redundant or noisy evidence and occasionally causes topic drift, leading to diminishing and sometimes negative returns.
>
> >**Q2:** Do you encounter issues with the shared memory or context growing unboundedly during long reasoning sequences? If so, how is this mitigated, and what is the computational or memory cost of maintaining such context?
>
> In principle, an agent’s shared context can grow if it keeps browsing without concluding. In practice this is rare in WebWatcher. We observed that training without SFT and directly using RL makes this phenomenon much more severe, while SFT cold-start followed by RL largely suppresses it.
>
> **Mitigations we use**:
> 1. We set the maximum trajectory length to 15, so the model learns not to produce unnecessarily long traces (Page 22, Line 1164-1168).
> 2. Text search keeps only Top-10 snippets and image search keeps Top-5 results. Both of them are ranked by relevance (Page 22, Line 1134-1147).
> 3. The visit tool writes a short summary to avoid consuming many tokens, instead of raw page dumps (Page 22, Line 1147-1156).
>
> We extend the context window to 128k tokens and explicitly instruct the model to output the final answer by round 15. The KV-cache is usually near 1 GB for WebWatcher-7B. We believe the most effective next step is to add a summary over the tool-call history itself, and we plan to continue optimizing in this direction.
>
> >**Q3:** It would also be helpful if the authors could provide qualitative visualizations comparing WebWatcher’s reasoning paths to those of baseline systems to highlight its hierarchical and multimodal advantages.
>
> Thank you for your suggestion. We add a case from HLE to Appendix F (Page 26-28, Line 1350-1481), where GPT-4o uses the same tools as WebWatcher. It shows that GPT-4o struggles to use the tools effectively in this case, repeatedly calling image search while performing image patch reasoning, and ultimately providing the wrong answer. In contrast, WebWatcher uses the tools proficiently and gives the correct answer (Page 28-30, Line 1482-1616).
>
> ---
> **Reference**
>
> [1] Wu J, Li B, Fang R, et al. Webdancer: Towards autonomous information seeking agency[J]. arXiv preprint arXiv:2505.22648, 2025.
>
> [2] Li K, Zhang Z, Yin H, et al. WebSailor: Navigating Super-human Reasoning for Web Agent[J]. arXiv preprint arXiv:2507.02592, 2025.
>
> [3] Luo J, Wu B, Luo X, et al. A survey on efficient large language model training: From data-centric perspectives. ACL 2025.
>
> [4] Wu W, Guan X, Huang S, et al. Wu W, Guan X, Huang S, et al. MASKSEARCH: A Universal Pre-Training Framework to Enhance Agentic Search Capability[J]. arXiv preprint arXiv:2505.20285, 2025.

---

> > ### Comment · Reviewer_cUPf · 2025-11-24
> > **Thanks for the response**
> >
> > The author response has addressed most of my concerns, and the additional fine-grained performance analysis is compelling. I will retain my score recommending acceptance.

---

> ### Author Response · Authors · 2025-11-25
> **Response to Reviewer cUPf**
>
> Dear Reviewer cUPf,
>
> Thank you for your response, and we are happy to see that our response address your concerns. Thank you again!

---

### Official Review · Reviewer_xaDU · 2025-11-01

**Soundness:** 2
**Presentation:** 3
**Contribution:** 2
**Rating:** 6
**Confidence:** 3

**Summary:**

The paper introduces WebWatcher, a multimodal deep research web agent designed to perform complex reasoning across both visual and textual information. It combines large language models with multiple external tools to handle information-seeking tasks that require cross-modal understanding and planning. The authors propose a new benchmark, BrowseComp-VL, extending previous text-based benchmarks into the visual domain. They construct synthetic multimodal question-answering dataset and use a two-stage training process: SFT and GRPO. Experiments show that WebWatcher achieves competitive performance on several benchmarks including HLE, LiveVQA, and MMSearch.

**Strengths:**

* The paper addresses an important area—multimodal deep research—by trajectory data creation and two-stage training process including SFT and RL.
* The proposed automated trajectory generation pipeline offers a scalable way to construct training samples for multi-model deep research.

**Weaknesses:**

* Although the paper mentions that the proposed benchmark was verified by PhD-level experts at line 144, it does not provide details on the verification process or quantitative reliability measures (e.g., Cohen’s κ), making the evaluation reliability insufficient.
* Several key baselines are missing, such as o3 and GPT-4.1, which limits the completeness and fairness of the performance comparison.

**Questions:**

* What are the performance results of o3 and GPT-4.1 in Table 1 and Table 2?
* Please provide more details about the manual verification process during the benchmark construction phase, including how the PhD-level experts conducted validation and whether any quantitative reliability metrics (e.g., inter-rater agreement) were reported.

---

> ### Author Response · Authors · 2025-11-21
> **Response to Reviewer xaDU**
>
> Thank you for your valuable feedback to help us improve our paper. We have revised our paper based on your feedback. We detail our response below and please kindly let us know if our response addresses your concerns.
>
> > **W1, Q2:** Although the paper mentions that the proposed benchmark was verified by PhD-level experts at line 144, it does not provide details on the verification process or quantitative reliability measures (e.g., Cohen’s κ), making the evaluation reliability insufficient. Please provide more details about the manual verification process during the benchmark construction phase, including how the PhD-level experts conducted validation and whether any quantitative reliability metrics (e.g., inter-rater agreement) were reported.
>
> We agree that providing details of the verification process and quantitative metrics is crucial for demonstrating our benchmark's reliability. To address your concern, we employed a dual independent verification and arbitration mechanism. Each benchmark sample (QA pair) was randomly assigned to two PhD-level experts for independent, blind evaluation, where they strictly verified the question quality, answer accuracy, and QA consistency. If the two experts' judgments were consistent, the sample was approved. If their opinions differed, the sample was escalated to a senior expert for a final, binding decision. To quantitatively assess reliability, we calculated the Inter-Rater Reliability (IRR). Before arbitration, the Initial Agreement Rate between the two independent experts was 89.3%. To account for chance agreement, we further calculated Cohen's Kappa ($\kappa$), which reached 0.86. According to established standards [1], this value indicates "Almost Perfect Agreement." We have added these details about the verification process and the $\kappa$ metric to Appendix E.9 (Page 25, Line 1341-1349) in the revised manuscript.
>
>
> > **W2, Q1:** Several key baselines are missing, such as o3 and GPT-4.1, which limits the completeness and fairness of the performance comparison. What are the performance results of o3 and GPT-4.1 in Table 1 and Table 2?
>
> Thank you for the valuable suggestion. Our primary comparison focuses on a broad range of open-source models and several representative closed-source systems, which we believe provides a comprehensive view of the current landscape. Nevertheless, we agree that including o3 and GPT‑4.1 can further strengthen the completeness of the evaluation. We add their performance results to Table R1 and the revised paper.
>
> **Table R1: Performance of o3 and GPT‑4.1 compared with WebWatcher-32B**
> | Model                | BC-VL L1 | BC-VL L2 | LiveVQA | MMSearch | SimpleVQA | HLE(avg.) |
> |----------------------|----------|----------|---------|----------|-----------|-----------|
> | o3                   | 26.7        | 23.0        | 50.0    | 54.3     | 70.3      | 21.4      |
> | gpt-4.1              | 15.6     | 6.0      | 32.3    | 26.0     | 60.3      | 11.2      |
> | **WebWatcher-32B (ours)** | 28.4     | 25.0     | 58.7    | 55.3     | 59.0      | 13.6      |
>
> ---
> **Reference**
>
> [1] Landis, J. Richard, and Gary G. Koch. "An application of hierarchical kappa-type statistics in the assessment of majority agreement among multiple observers." Biometrics (1977): 363-374.

---

> ### Author Response · Authors · 2025-11-26
> **We would like to hear back from reviewer xaDU**
>
> Dear reviewer xaDU,
>
> We would like to follow up to see if the response addresses your concerns. We would really appreciate the opportunity to discuss this further if our response has not already addressed your concerns. Thank you again!

---

### Author Response · Authors · 2025-11-21
**Summary of Paper Revision**

We sincerely appreciate all reviewers for their insightful and constructive feedback. According to these comments, we have improved the paper (new pdf uploaded) and highlighted the main changes with blue text. Below, we summarize all changes:

---
**Reviewer xaDU**
1. We have added details about the verification process and the $\kappa$ metric to Appendix E.9 (Page 25, Line 1341-1349).
2. We have supplemented the performance of o3 and GPT‑4.1 in Tab. 1 and Tab.2 (Page 6-7).
---
**Reviewer cUPf**

3. We have compiled resource efficiency metrics for WebWatcher-7B, and add them to Appendix E.6 (Page 24, Line 1249-1277).
4. We have added the future plan of scaling up trajectory annotation with open-source models to Sec. 5 (Page 10, Line 536-539).
5. We have included an additional case from HLE conducted by GPT-4o as a comparative example in Appendix F (Page 26-28, Line 1350-1481).
---
**Reviewer ehqj**

6. We have revised the claim of performance in Abstract (Line 23-26, Page 1).
7. We have conducted an ablation study of the number of tool calls in Sec. 4.3 (Page 8, 428-442).
8. We have reported a human baseline on L1/L2 in Sec. 4.4 (Page 9, Line 432-450).
9. We have added a table of error distribution and its analysis to Appendix E.8 (Page 24, Line 1292-1340).
10. We have included a table of inter-rater agreement between LLM judges and human experts and its analysis to Appendix E.7 (Page 24, Line 1285-1304).
11. We have clarified the cost-latency of Pass@k in Appendix E.6 (Page 24, Line 1772-1884).
---
**Reviewer sPg8**

12. We have reported a clarification about entities in the input image and visual difficulty in Sec. 2.3 (Page 5, Line 218-222).
13. We have revised the reasoning description in Abstract (Page 1, Line 17-18).

---

### Author Response · Authors · 2025-11-30
**Rebuttal Summary for AC**

Dear Program Chairs, Senior Area Chairs, Area Chairs and Reviewers:

We sincerely thank you for your time and expertise, and we are grateful to all reviewers — xaDU, cUPf, ehqj, and sPg8 — for their constructive insights.
During the rebuttal period, we provided detailed, point-by-point responses to all reviewers and added comprehensive supplementary experiments addressing every raised concern. Reviewer cUPf acknowledged that our rebuttal satisfactorily resolved all of their issues and explicitly **recommended acceptance**. Reviewer sPg8 engaged with us through two rounds of discussion, and we have fully clarified and answered all of their follow-up questions. Although the remaining two reviewers did not respond further, we nonetheless addressed all of their comments thoroughly and included substantial additional experiments that directly target their concerns.

We fully understand that this is a particularly busy period for ACs. We hope these actual outcomes of the rebuttal process will be fully considered in your final decision-making. Deeply thank you again for your time and for handling this challenging situation.

---

### Meta-Review · Area_Chair_98Ag · 2026-01-06

**Summary:**

The paper introduces WebWatcher, a multimodal web-research agent built for information-seeking problems that require combining visual and textual evidence through multi-step tool use. In addition, the authors propose BrowseComp-VL, a vision-language extension of BrowseComp for multi-hop retrieval, evidence aggregation, and cross-modal grounding. WebWatcher achieves strong performance on several high-difficulty benchmarks, including HLE, LiveVQA, BrowseComp-VL, and MMSearch.

The main concerns from reviewers (scores: 6 / 8 / 4 / 4) for the original version are:

1. Improper performance claims and missing key baselines (e.g., o3, GPT-4.1) (ehqj, xaDU)

2. Benchmark reliability and human alignment (xaDU, ehqj)

3. Potential inductive bias in trajectory filtering (ehqj)

4. Training and inference cost were not clearly reported (cUPf)

**Reviewer Concerns:**

Reviewers generally agreed that the problem is timely and that the proposed system and benchmark are meaningful for multimodal deep-research agents.

The rebuttal and revision also improved clarity and rigor by addressing the main evaluation and reporting gaps, strengthening experimental coverage and analyses, and clarifying key claims and design choices.

Overall, these updates resolved all the reviewer concerns and were strong enough that Reviewer cUPf explicitly maintained an accept score and noted that the response addressed the majority of issues.

**Reviewer Scores:**

xaDU (6): likely keep 6, since missing baselines and verification details were addressed directly.

cUPf (8): confirmed acceptance and retained score.

ehqj (4): may raise to 6. The major concerns were responded to with claim revision, tool-call ablation, human baseline, judge reliability, and cost–latency.

sPg8 (4): may raise to 6 or may not change. The main concern about multimodality contributions and isolating visual components is well explained in the rebuttal, and the MMSearch-Plus comparison given in the rebuttal may convince the reviewer to raise the score.

---

### Decision · Program_Chairs · 2026-01-26

Accept (Poster)